# Pharmacological Screening of Kv7.1 and Kv7.1/KCNE1 Activators as Potential Antiarrhythmic Drugs in the Zebrafish Heart

**DOI:** 10.3390/ijms241512092

**Published:** 2023-07-28

**Authors:** Alicia De la Cruz, Xiaoan Wu, Quinn C. Rainer, Irene Hiniesto-Iñigo, Marta E. Perez, Isak Edler, Sara I. Liin, H. Peter Larsson

**Affiliations:** 1Department of Physiology and Biophysics, Miller School of Medicine, University of Miami, 1600 NW 10th Avenue, Miami, FL 33136, USA; 2Department of Biomedical and Clinical Sciences, Linköping University, SE-581 85 Linköping, Sweden

**Keywords:** *I*
_Ks_, long QT syndrome, polyunsaturated fatty acids, zebrafish heart, ML-277

## Abstract

Long QT syndrome (LQTS) can lead to ventricular arrhythmia and sudden cardiac death. The most common congenital cause of LQTS is mutations in the channel subunits generating the cardiac potassium current *I*_Ks_. Zebrafish (*Danio rerio*) have been proposed as a powerful system to model human cardiac diseases due to the similar electrical properties of the zebrafish heart and the human heart. We used high-resolution all-optical electrophysiology on ex vivo zebrafish hearts to assess the effects of *I*_Ks_ analogues on the cardiac action potential. We found that chromanol 293B (an *I*_Ks_ inhibitor) prolonged the action potential duration (APD) in the presence of E4031 (an *I*_Kr_ inhibitor applied to drug-induced LQT2), and to a lesser extent, in the absence of E4031. Moreover, we showed that PUFA analogues slightly shortened the APD of the zebrafish heart. However, PUFA analogues failed to reverse the APD prolongation in drug-induced LQT2. However, a more potent *I*_Ks_ activator, ML-277, partially reversed the APD prolongation in drug-induced LQT2 zebrafish hearts. Our results suggest that *I*_Ks_ plays a limited role in ventricular repolarizations in the zebrafish heart under resting conditions, although it plays a more important role when the *I*_Kr_ is compromised, as if the *I*_Ks_ in zebrafish serves as a repolarization reserve as in human hearts. This study shows that potent *I*_Ks_ activators can restore the action potential duration in drug-induced LQT2 in the zebrafish heart.

## 1. Introduction

Long QT syndrome (LQTS) is a condition that delays the repolarization of the heart, predisposing individuals to arrhythmias, which can result in syncope, seizures, or sudden death [1,2]. LQTS affects an estimated 1 in 2000–7000 people and is defined by a prolonged corrected QT interval (QTc) reading on a surface electrocardiogram (ECG) [3]. The QT interval is a surrogate measure of the ventricular action potential duration (APD). The causes of LQTS include both congenital and acquired alterations in certain cardiac ion channels. These alterations can cause abnormal ion currents within the heart and thus increase the QT interval and the APD. We here tested whether zebrafish hearts would be a good model to study LQTS and for the development of pharmacological treatments for LQTS.

Zebrafish have been suggested as a model organism for studying LQTS [4]. Despite distinct differences from mammalian hearts, such as the presence of two rather than four cardiac chambers, the zebrafish heart is remarkably similar to mammalian hearts in numerous ways, which makes it suitable as a simple model system. The electrophysiology of the zebrafish heart has been thoroughly studied, demonstrating pace-making activity and a ventricular APD resembling that of the human heart [5,6]. Furthermore, larval and adult zebrafish ECG recordings demonstrate a similar sequence of electrical activation and relaxation as seen in human hearts [7]. Moreover, zebrafish ECG recordings display comparable resting heart rates and conduction intervals to human ECG recordings, and the QT interval has a nearly linear relationship with the RR interval [8,9]. This allows for the development of zebrafish models that can be used to examine delayed ventricular repolarization and prolongations of the APD and QTc interval (i.e., LQTS) and to develop pharmacological interventions to reverse these disorders. 

There are two major delayed rectifier currents in the ventricles of the human heart, the *I*_Kr_ and *I*_Ks_ [10]. The *I*_Kr_ is generated by the Kv11.1 channel and is the rapid delayed rectifier current [11]. The slow delayed rectifier current, the *I*_Ks_, is generated by channels consisting of tetramers of the Kv7.1 (KCNQ1) α-subunit associated with ancillary KCNE1 β-subunits. At resting heart rates, most of the repolarizing potassium current in human ventricles is generated by Kv11.1 channels, while Kv7.1/KCNE1 channels act as a reserve when additional repolarizing current is needed, typically during high sympathetic tone [8,12]. The repolarization reserve capabilities of K+ currents are utilized as a mechanism to protect against cardiac arrhythmias in both physiological and pathological states [13]. There have been multiple genes identified which encode components of such ion channels, and mutations in these genes result in specific subtypes of LQTS. For instance, LQT1 involves a mutation in the *KCNQ1* gene, LQT2 involves a mutation in the *KCNH2* gene (which encodes the Kv11.1 channel), and LQT5 involves a mutation in the *KCNE1* gene [14].

Given the roles of the *I*_Kr_ and *I*_Ks_ in the pathophysiology of LQTS in humans, quantifying the prevalence of potassium channels such as Kv7.1 and Kv11.1 in zebrafish hearts is of paramount importance for assessing their use in cardiac models. Previous studies utilizing E4031, a selective Kv11.1 channel blocker, have consistently demonstrated a prolongation of the APD in both the ventricle and atrium of zebrafish hearts, indicating that the *I*_Kr_ plays a significant role in the action potential plateau duration of the zebrafish heart [6,15]. On the contrary, previous studies utilizing chromanol 293B, a Kv7.1 and Kv7.1/KCNE1 channel blocker, did not demonstrate a significant effect on the ventricular APD in 48-hour-old zebrafish embryos [16]. Additionally, immunohistochemistry staining of the embryonic heart of a zebrafish revealed no Kv7.1 [16]. These findings suggest that embryonic zebrafish hearts do not possess the *I*_Ks_. However, transcripts of the KCNQ1 α-subunit have been detected in the adult zebrafish heart, and chromanol 293B was reported to prolong the QT interval and ventricular APD in adult zebrafish [17,18,19]. 

The difficulty of functionally identifying the *I*_Ks_ within zebrafish cardiomyocytes may be because the *I*_Ks_ is masked under a larger *I*_Kr_ and only becomes apparent under conditions which minimize the *I*_Kr_ or when additional K^+^ currents are needed. This is similar to how the *I*_Ks_ acts as a repolarization reserve in some mammalian hearts. Therefore, one of the purposes of the work presented here was to examine the roles of the *I*_Ks_ in zebrafish hearts under rest and stress (drug-induced LQT2). These were assessed by measuring changes in the APD after applying the *I*_Ks_ blocker chromanol 293B alone and after applying chromanol 293B for further APD prolongation following *I*_Kr_ block with E4031, respectively. In addition, the same approach was performed with different *I*_Ks_ activators.

This work also evaluated the zebrafish heart as a model for the study of the effects of polyunsaturated fatty acid (PUFA) analogues on the APD. As of late, PUFAs are being studied as a potential medical therapy for treating diseases such as LQTS. The treatment of LQTS currently relies primarily on beta-blockers, flecainide (NaV channel inhibitor), implanted cardioverter defibrillators, and sympathetic denervation [20,21,22,23,24,25]. None of these treatments address the pathological process of LQTS and thus do not restore the QT interval, but rather prevent or stop dangerous cardiac arrhythmias [26]. Furthermore, some patients are unable to tolerate certain treatments due to the side effects. A recent simulation study suggested that enhancement of the *I*_Ks_, which can be achieved using certain PUFAs and PUFA analogues, is a potential safe and effective LQTS treatment with antiarrhythmic effects [27,28].

The chemical structures of PUFAs consist of a long hydrocarbon tail with two or more double bonds attached to a charged, hydrophilic head group [29] (Figure 1A). The mechanism by which PUFAs enhance ion currents is referred to as the lipoelectric hypothesis, in which the PUFAs incorporate into the cell membrane adjacent to an ion channel [30]. From there, a PUFA’s headgroup interacts with positive charges in the voltage-sensing domain to negative shift the V_50_ of activation, and in the pore domain, to increase the conductance of the Kv7.1/KCNE1 [31] (Figure 1A,B). While PUFAs can affect cardiac NaV, CaV, and Kv channels, their selectively and effect depend on the specific PUFA analogues [2]. Some PUFA analogues, such as docosahexaenoyl glycine (DHA-Gly), have demonstrated QT-interval shortening effects with a pronounced increased activity of Kv7.1/KCNE1 within ex vivo and in vivo guinea pig hearts as well as shortening the APD of human-induced pluripotent stem-cell derived cardiomyocytes [2,32]. The work presented here also analyzed the effect of DHA-Gly and linoleoyl glycine (LIN-Gly), two PUFA analogues that were previously found to be more selective for human *I*_Ks_ channels compared to other PUFA analogues [2], on zebrafish hearts ex vivo. Our hypothesis was that the zebrafish heart ex vivo might be a good model for the early screening of Kv7.1 and *I*_Ks_ activators for their effect on the cardiac action potential related to its effects previously described on Kv7.1 and the *I*_Ks_. 

We found that the PUFA analogues tested only slightly affected the zebrafish APD. However, a stronger Kv7.1 activator (ML-277) shortened the zebrafish APD, which shows that zebrafish represent a possible model for testing antiarrhythmic drugs related to Kv7.1 and Kv7.1/KCNE1 when using activators with larger effects and great affinity.

## 2. Results

### 2.1. AP Optical Recordings Are Stable in Control Conditions

The APD of the zebrafish ventricle was measured optically under control conditions at different percentages of repolarization (APD25, APD50, APD80, and APD90) (Figure 1C, left), and the change in ΔAPD (as a percentage) was calculated between the control conditions and the drug or the PUFA analogues conditions (Figure 1C, right). The APD80 average (in ms) for the control conditions was 211.8 ± 8.3 ms (n = 54). The APD80 (in ms) for the control conditions was also analyzed at different recording times as time-matched controls to rule out effects not caused by the drugs (Figure 1D). No significant differences were found in the APD80 for the control conditions at 10, 20, or 30 min after the first control condition recording, nor when the average of the APD80 measured at the three times was compared with the first control (Figure 1D, n = 4, NS, *p* > 0.05, Table 1). 

### 2.2. The I_Kr_ Inhibitor E4031 Prolongs Zebrafish APD

We started by analyzing the effects of E4031, an *I*_Kr_ inhibitor. 1 µM E4031 was previously reported to prolong the zebrafish ventricle APD90 by ≈24% [6]. We used E4031 as a control and to generate drug-induced LQT2 zebrafish hearts. Figure 2A shows representative traces of the AP before (black) and after (red) 30 min of exposure to E4031 (10 µM). E4031 (10 µM) prolonged the APD80 by 72.8 ± 5.7% (n = 26, *p* < 0.001). The drug effect on the APD80 (ΔAPD80%) was measured at 10, 20, 30 min after applying the drug (Figure 2B). E4031 (10 µM) significantly increased the APD80 at all the times recorded (*vs.* the control condition APD80), and no significant differences were observed among the different times recorded after applying the drug. Because E4031 was used for the drug-induced LQT2 condition, a control with this drug was also measured after an additional 10, 20, 30 min after being applied for a second time to analyze whether the effect of the E4031 was stable after the first 30 min (Figure 2B, Table 1). 30 min after applying E4031, the APD prolongation remained stable and no increase in the APD80 was observed (Table 1). To further analyze the effect of E4031, we examined the drug effect at different percentages of AP repolarization at 30 min after application. We measured the ΔAPD (%) for the APD25, APD50, APD80, and APD90 (Figure 2C). E4031 (10 µM) increased the APD at every percentage (Table 2). However, the most significant change observed was at the APD80 and APD90, as expected for a Kv11.1 channel blocker (Table 2). We also analyzed the AP repolarization during phase 3, specifically because it is the AP phase where the *I*_Kr_ and *I*_Ks_ are mainly involved in the AP. Therefore, to analyze phase 3, we calculated the difference between the APD80 and APD25 (in ms) and compared the drug condition vs. the control condition values. Phase 3 was significantly affected by 10 µM E4031 (233.7 ± 10.0 ms vs. 113.4 ± 4.3 ms for the APD80–APD25 in the E4031 and control conditions, respectively, n = 26, *p* < 0.001) (Figure 2D). E4031 prolonged the zebrafish AP under all the conditions tested, demonstrating that the *I*_Kr_ contributes to the repolarization of the ventricular AP in zebrafish, which is consistent with previous studies in the literature [6]. 

### 2.3. Two Different Human I_Ks_ Inhibitors Have Different Effects on Zebrafish AP

Chromanol 293B is a human Kv7.1/KCNE1 selective channel blocker with an IC50 of ≈7 µM [33]. Chromanol 293B (100 µM) significantly prolonged the zebrafish APD80 by 9.6 ± 0.3% at 10 min after application (n = 3, *p* < 0.05) (Figure 2E,F). This small but significant effect in the APD80 was not found at 20 or 30 min after applying chromanol 293B, nor when the average of the APD80 values at different times was compared with the control condition values (Figure 2F, Table 1). However, this small but significant effect at 10 min was similar to the one observed by Abramochlin et al. in isolated zebrafish cardiomyocytes [19]. When we examined the ΔAPD at different percentages of repolarization (at T = 10 min), we observed that chromanol 293B significantly prolonged the APD80 and APD90 (Figure 2G, Table 2). Furthermore, chromanol 293B modified phase 3 of the AP (144.6 ± 15.0 ms vs. 118.3 ± 12.6 ms for APD80-APD25 in chromanol 293B and control conditions, respectively, n = 3, *p* < 0.05) (Figure 2H), as expected for an *I*_Ks_ inhibitor. 

Because the effects of chromanol 293B were small, we studied another potent human Kv7.1/KCNE1 channel blocker, JNJ303. The IC50 for JNJ303 was previously found to be 78 nM for human Kv7.1/KCNE1 [34]. After 30 min, JNJ303 (10 µM) prolonged the APD80 by 16.4 ± 3.4% (n = 3, *p* < 0.01, Figure 2J). Surprisingly, JNJ303 showed a greater effect on the APD25 than on the APD80 or APD90 (Figure 2K, Table 2), suggesting that JNJ303 might be affecting other ion channels involved in the zebrafish AP. Moreover, JNJ303 did not significantly affect phase 3 of the zebrafish AP (Figure 2L, 92.7 ± 5.5 ms vs. 88.2 ± 7.9 ms for APD80–APD25 in JNJ303 and control conditions, respectively, n = 3, NS, *p* > 0.05), as would have been expected if JNJ303 inhibited the *I*_Ks_ channels.

### 2.4. The I_Ks_ Blocker Chromanol 293B Prolongs the APD More after I_Kr_ Inhibition

In humans, the *I*_Kr_ is the primary contributor current for AP repolarization. However, mutations in the Kv7.1/KCNE1 channel, which generates the *I*_Ks_, can cause fatal cardiac arrhythmia. This suggests that the *I*_Ks_ is also crucial for repolarization, although its importance depends on the underlying conditions. For instance, Kv7.1/KCNE1 channel activation is essential for the fight-or-flight response of the sympathetic nervous system. In addition, Kv7.1/KCNE1 channel activation has been described as crucial in stressful situations and in any exercise activity where the heart rate is increased. Therefore, we examined the contribution of the *I*_Ks_ when the *I*_Kr_ was compromised to determine if the *I*_Ks_ is more significant in the zebrafish heart under this condition.

To evaluate the *I*_Ks_ contribution when the *I*_Kr_ is compromised, we first recorded the zebrafish AP in the absence and the presence of E4031. We then waited 30 min after the application of E4031 before we applied a combination of E4031 plus chromanol 293B. Figure 3A shows representative traces before applying E4031 (black), after applying E4031 (10 µM, red), and after applying E4031/chromanol 293B (10 µM/100 µM, dark purple). In the presence of E4031/chromanol 293B (10 µM/100 µM), the APD80 was significantly prolonged by 20.2 ± 2.0% compared to the recording before the application of the combination of drugs (n = 3, *p* < 0.01). In the control condition with only E4031, the APD80 reached a steady state 30 min after applying the drug and the APD80 did not prolong after that (Figure 2B, Table 1). After applying E4031/chromanol 293B (10 µM/100 µM), we observed a significant APD80 prolongation after 30 min (Figure 3B), and when the APD80 values measured at different times (10, 20, 30) were averaged and compared with the E4031 APD80 values (Table 1). Moreover, when we analyzed the APD at different repolarization percentages, we observed that E4031/chromanol 293B generated a significant APD prolongation for the APD80 and APD90 compared with its control E4031 alone (Figure 3C, Table 2). We observed that chromanol 293B prolongation affected AP phase 3 (Figure 3D), as expected for a Kv7.1/KCNE1 channel blocker (264.3 ± 19.24 ms vs. 317.2 ± 12.7 ms for APD80-APD25 in E4031 conditions and E4031/chromanol 293B, respectively, n = 3, *p* < 0.05). 

Our results show that chromanol 293B has a larger effect on the APD when the *I*_Kr_ is blocked, suggesting that the *I*_Ks_ is more important for repolarization in zebrafish hearts when the *I*_Kr_ is compromised. 

### 2.5. The Human I_Ks_ Inhibitor JNJ303 Does Not Prolong the APD More after I_Kr_ Inhibition

To test further whether JNJ303 affects Kv7.1/KCNE1 in the zebrafish heart, we combined E4031 and JNJ303. In the presence of E4031/JNJ303 (10 µM/10 µM), the APD80 was significantly prolonged by 14.3 ± 4.3% compared to the recording before the application of the combination of drugs (n = 5, *p* < 0.05) (Figure 3E). We observed additional APD80 prolongation at all the times tested after applying E4031/JNJ303 (Figure 3F). Surprisingly, when we analyzed the E4031/JNJ303 combination at different percentages of repolarization (Figure 3G), we observed that the longest AP prolongation occurred at the APD25 and APD50 instead of at the APD80 or APD90 (Table 2), as would be expected for a compound that affects Kv7.1/KCNE1. Moreover, the combination of E4031/JNJ303 did not significantly affect phase 3 of the zebrafish AP (Figure 3H, 231.5 ± 23.5 ms vs. 214.3 ± 16.2 ms for APD80–APD25 in E4031/JNJ303 and E4031 conditions, respectively, n = 5, NS, *p* > 0.05). These results are similar to what we observed for JNJ303 alone and suggest that JNJ303 does not inhibit the zebrafish *I*_Ks_ to a large extent. 

### 2.6. JNJ303 Is Not an Effective I_Ks_ Inhibitor of Zebrafish Kv7.1/KCNE1 Currents

To test directly the effect of JNJ303 on zebrafish Kv7.1/KCNE1, we expressed zebrafish Kv7.1/KCNE1 in *Xenopus laevis* oocytes and applied JNJ303 and chromanol 293B (Appendix A and S2, respectively). Only the application of very high concentrations of JNJ303 inhibited the zebrafish Kv7.1/KCNE1 currents (Appendix A). The IC50 value estimated for JNJ303 for zebrafish Kv7.1/KCNE1 was >100 µM (n = 3) (Appendix A), much higher than what was described for human Kv7.1/KCNE1 (IC50 ≈ 78 nM) [34]. Together with our results concerning ex vivo hearts for JNJ303, these results suggest that JNJ303 might be affecting different cardiac ion channels involved in the zebrafish AP than Kv7.1/KCNE1. 

To confirm that chromanol 293B directly affects zebrafish Kv7.1/KCNE1, we used the same approach for zebrafish Kv7.1/KCNE1 expressed in *Xenopus laevis* oocytes with chromanol 293B (Appendix A). We also tested chromanol 293B in zebrafish Kv7.1. Here, 10 µM chromanol 293B inhibited the zebrafish Kv7.1/KCNE1 currents (Appendix A) by 43.7 ± 0.04% (n = 5, *p* < 0.01, Appendix A). At the concentration tested, chromanol 293B shifted the voltage dependence of the activation of zebrafish Kv7.1/KCNE1 slightly toward positive potentials (n = 5, *p* < 0.05). Moreover, 10 µM chromanol 293B inhibited the zebrafish Kv7.1 current (Appendix A) by 76.0% ± 0.04% (n = 5, *p* < 0.001, Appendix A). In human Kv7.1/KCNE1, the chromanol 293B IC50 (≈7 µM) was described by Lerche et al. in 2007. The inhibition induced in human Kv7.1/KCNE1 was more potent than in the human Kv7.1 homomeric channel (IC50 ≈ 27 µM) [33]. Our results suggest that the inhibition of zebrafish Kv7.1/KCNE1 was similar to that observed in human Kv7.1/KCNE1. However, the chromanol 293B effect observed on zebrafish Kv7.1 was greater than that seen in zebrafish Kv7.1/KCNE1.

### 2.7. PUFA Analogues Have Modest Effects on Zebrafish AP

To determine whether the zebrafish heart is an appropriate model for early screening of human antiarrhythmic drugs affecting the *I*_Ks_ channels, we examined the effects of PUFA analogues on the optical recordings of the zebrafish AP. DHA-Gly (20 µM) shortened the APD80 by 6.1 ± 2.3% (n = 9, *p* < 0.05) at 30 min (Figure 4A). However, DHA-Gly did not significantly shorten the APD80 when averaging the APD80 values at different times (10, 20, and 30 min) and comparing the result to the control condition (Table 1). We also analyzed the DHA-Gly effect at different percentages of repolarization (Table 3). DHA-Gly only shortened the APD80 (Figure 4C, Table 3). However, when we analyzed phase 3 of the AP (Figure 4D), we did not find any significant difference between the control conditions and after applying the PUFA analogue (108.1 ± 4.9 ms vs. 118.0 ± 6.8 ms for APD80-APD25 in DHA-Gly and control conditions, respectively, n = 9, NS, *p* > 0.05). 

LIN-Gly (20 µM) shortened the AP by 4.9 ± 0.8% (n = 6, *p* < 0.01) 20 min after application (Figure 4E,F). LIN-Gly shortened the APD80 at both 10 min (by 3.4 ± 0.3%, n = 5, *p* < 0.001) and at 20 min after applying the LIN-Gly (by −4.9 ± 0.8%, n = 6, *p* < 0.01). However, this small effect was not maintained over time and no AP shortening was seen after 30 min of exposure to the LIN-Gly (Figure 4F). However, LIN-Gly shortened the APD80 when the average of the APD80 values at the different times was compared with the control conditions’ APD80 values (Table 1). Figure 4G shows the ΔAPD measured at different percentages of repolarization. LIN-Gly significantly shortened the APD80, as expected for a compound that might be activating Kv7.1/KCNE1 (Table 3). We did not observe any significant change in phase 3 between the control conditions and the PUFA analogue conditions (Figure 4H, 112.4 ± 4.3 ms vs. 127.2 ± 1.7 ms for APD80-APD25 in LIN-Gly and control conditions, respectively, n = 6, NS, *p* > 0.05). These results indicate that PUFA analogues only slightly, or not significantly, shorten the zebrafish AP. 

### 2.8. PUFA Analogues Do Not Reverse APD in Drug-Induced LQTS Zebrafish Hearts

Because the *I*_Ks_ contribution to the repolarization seems more prominent when the *I*_Kr_ is compromised (Figure 3), we used the *I*_Kr_ channel blocker E4031 (i) to elucidate whether the PUFA analogues’ effects observed were due to targeting the *I*_Ks_ (i.e., we would expect a greater PUFA-induced shortening of the zebrafish AP in the presence of E4031) and (ii) to examine the ability of PUFA analogues to reverse the prolonged AP when LQT2 is drug-mimicked. To that end, we first applied E4031 and waited 30 min. We then recorded the zebrafish AP after applying E4031 combined with the PUFA analogues DHA-Gly or LIN-Gly. Figure 5A shows the zebrafish AP representative traces before the application of E4031 (black), after the application of E4031 (red), and after E4031/DHA-Gly (dark green). No significant effects were observed after applying E4031/DHA-Gly at any time tested when analyzed at the APD80 (Figure 5B, n = 4, NS, *p* > 0.05). Moreover, no significant effects were observed when analyzing the average of the APD80 values at different times (10, 20, 30) compared with the E4031 APD80 values (Table 1). However, when we analyzed the APD at different percentages of repolarization, surprisingly, we observed a significant prolongation of the APD90 (Figure 5C, Table 3). Moreover, when we analyzed phase 3, we observed a significant prolongation of phase 3 when comparing the combined compounds vs. the control condition E4031 alone (250.4 ± 35.1 ms vs. 199.2 ± 30.4 ms APD80–APD25 for E4031/DHA-Gly and E4031, respectively, n = 4, *p* < 0.05. Figure 5D). These results show that DHA-Gly did not shorten the APD when LQT2 was drug-induced. On the contrary, these results suggest that the combination of E4031 and DHA-Gly affects the zebrafish AP by increasing the APD. 

We used the same approach with the other PUFA analogue, LIN-Gly. Figure 5E shows representative traces before (black) and after applying E4031 (red) and then after applying E4031/LIN-Gly (purple). The combination of E4031/LIN-Gly (10 µM/20 µM) prolonged the APD80 compared to its control E4031 (10 µM) when recorded at 20 min. However, it did not promote any effect at 30 min after application (Figure 5F). No significant effects were observed when analyzing the average of the E4031/LIN-Gly APD80 values at different times (10, 20, 30) compared with the E4031 APD80 values (Table 1). The combination of E4031/LIN-Gly (10 µM/20 µM) also prolonged the APD25 (Figure 5G, Table 3). This result agrees with what was observed when analyzed in phase 3 of the AP (Figure 5H, 274.2 ± 34.6 ms vs. 273.4 ± 17.4 ms APD80–APD25 for E4031/LIN-Gly and E4031, respectively, n = 4, NS, *p* > 0.05). These results show that DHA-Gly and LIN-Gly do not shorten the APD in drug-induced LQT2 zebrafish hearts. 

### 2.9. DHA-Gly and LIN-Gly Have Only Modest Effects on Zebrafish Kv7.1/KCNE1 and Kv7.1 Channels

Because LIN-Gly and DHA-Gly did not shorten the APD in drug-induced LQT2 zebrafish hearts, we wanted to directly analyze the effects of DHA-Gly and LIN-Gly on zebrafish Kv7.1 and Kv7.1/KCNE1. Thus, the PUFA analogues were tested on the zebrafish Kv7.1/KCNE1 channel expressed in *Xenopus laevis* oocytes (Appendix A). Application of 20 µM DHA-Gly did not modify the current amplitude or the voltage dependence of zebrafish Kv7.1/KCNE1 (Appendix A, Appendix A). However, 20 µM LIN-Gly promoted the activation of the Kv7.1/KCNE1 by left-shifting the voltage-dependence of activation (ΔV_0.5_), and it increased the magnitude of the current (I/I0) of the zebrafish Kv7.1/KCNE1 (Appendix A, Appendix A) at a voltage (V = 0 mV) that mimics the voltage at the action potential plateau in zebrafish ventricles. Moreover, the PUFA analogues were tested on the zebrafish Kv7.1 expressed in *Xenopus laevis* oocytes (Appendix A). Application of 20 µM of DHA-Gly left-shifted the V0.5 by 12.2 ± 2.7 mV (n = 4, *p* < 0.05) and increased the maximal conductance of the zebrafish Kv7.1 (n = 4, *p* < 0.01). DHA-Gly increased the magnitude of the current (I/I0) of the zebrafish Kv7.1 (n = 4, *p* < 0.001) (Appendix A, Appendix A). LIN-Gly showed similar effects to DHA-Gly in zebrafish Kv7.1 (Appendix A, Appendix A). Moreover, LIN-Gly showed similar effects in zebrafish Kv7.1 and zebrafish Kv7.1/KCNE1. These results show that DHA-Gly and Lin-Gly only modestly (around two-fold), or not at all, increase the currents in Kv7.1/KCNE1 and Kv7.1. 

### 2.10. DHA-Gly and LIN-Gly Did Not Reverse the Triangulation Effect of Drug-Induced LQT2

Triangulation is an outcome of some drugs that cause LQTS [35]. For example, when APD25 or APD50 do not change by the drug, although the APD80 or APD90 is prolonged, the shape of the AP changes and it appears more similar to a triangle. This change in the AP shape can be quantified by analyzing and comparing the APD25/APD80 ratio between the control conditions and drug conditions. Figure 6A shows AP traces in the control conditions (black) and after applying the *I*_Kr_ inhibitor (red), the *I*_Ks_ inhibitor (blue and light blue), and the combination of the *I*_Kr_ and *I*_Ks_ inhibitors (dark purple and dark turquoise). The control condition showed a more squared AP, whereas the E4031, chromanol 293B, and the combination of E4031/chromanol 293B AP shapes were more triangular. However, the AP shapes in JNJ303 and the combination of E4031/JNJ303 were more similar to that observed in the control condition, i.e., a squared AP shape. E4031, chromanol 293B, and the combination of E4031/chromanol 293B significantly decreased the APD25/APD80 ratio (Figure 6B), corroborating what we observed in the qualitative analysis (Table 4). However, JNJ303 and the combination of E4031/JNJ303 did not change the AP shape or reduce the APD25/APD80 ratio compared to the control conditions (Table 4). For the PUFA analogues and their combination with E4031 (Figure 6C,D), we expected a more squared AP shape, even more so than the control conditions (Figure 6A), and an increase in the APD25/APD80 ratio due to the expected increase in the *I*_Ks_. However, there was no change between the control conditions and the PUFA analogue conditions, neither qualitative nor quantitative (Figure 6C,D, Table 4). Moreover, when we analyzed the E4031/PUFA analogues combination, we found that the PUFA analogues did not reverse the triangulation promoted by E4031 (Figure 6C,D).

### 2.11. The I_Ks_ Activator ML-277 Partially Restored the APD in Drug-Induced LQT2 Zebrafish Hearts

The PUFA analogues caused only a small shortening of the zebrafish APD. Hence, we wanted to test whether a stronger Kv7.1 activator would shorten the zebrafish APD more. We therefore examined the effects of ML-277 (IC50 = 270 nM for human Kv7.1) on the zebrafish APD. ML-277 was first tested on zebrafish Kv7.1 expressed in *Xenopus laevis* oocytes, where it increased the current substantially more (>8 fold) than the PUFA analogues (Appendix A, Appendix A). ML-277 (10 µM) shortened the APD80 after 10, 20 and 30 min of application (Figure 7A,B). Moreover, significant APD shortening was observed when the average of the APD80 values at the different times (10, 20, 30) was compared with the APD80 values in control conditions (Table 1). Moreover, ML-277 shortened the APD at different percentages of repolarization, APD50 and APD80 (Figure 7C, Table 2). However, ML-277 did not show any effect on phase 3 of the repolarization (Figure 7D). To determine whether, as for chromanol 293B, the effect of ML-277 might be increased when the *I*_Kr_ channels were compromised, we examined the effect of ML-277 on the zebrafish APD in drug-induced LQT2. We therefore applied E4031(10 µM) and E4031/ML-277(10 µM/10 µM) sequentially. ML-277 shortened the APD80 by 13.6 ± 2.6% compared with its control E4031 alone (Figure 7E, n = 5, *p* < 0.01). After applying E4031/ML-277, the APD80 showed significant shortening at all the times tested (Figure 7F) when we compared it with its control with E4031 suggesting that ML-277 partly restored the APD80 when drug-induced LQT2 (Figure 7F). Moreover, ML-277 significantly shortened the APD when the average of the APD80 values of the different times (10, 20, 30) was compared with the APD80 values in control conditions (Table 1). Moreover, ML-277 significantly shortened the APD50, APD80 and APD90 (Figure 7G, Table 2). ML-277 also significantly shortened phase 3 of the repolarization in drug-induce LQT2 (Figure 7H, 164.4 ± 21.4 vs. 208.7 ± 16.7 APD80–APD25 for E4031/ML-277 and E4031, respectively, n = 5 *p* < 0.05). Although there was not a considerable increase in the ML-277 change at the APD80 (%) in the presence of E4031 compared to the absence of E4031, these results still suggest that the effect of ML-277 on the APD is greater when *I*_Kr_ is compromised because ML-277 also significantly affected the APD90 and phase 3 of the repolarization, while these ML-277 effects were not observed in the absence of E4031.

## 3. Discussion

This study evaluated the zebrafish heart as a model for early screening of the *I*_Ks_ activators as potential human antiarrhythmic drugs. We drug-induced LQTS in zebrafish hearts with the application of chromanol 293B (*I*_Ks_ blocker) and E4031 (*I*_Kr_ blocker) to simulate LQT1/5 and LQT2, respectively. E4031 was found to significantly prolong the zebrafish heart APD, as has been previously described [6], although we here tried a higher concentration (10 vs. 1 μM in the previous study) and found additional APD prolongation at this higher concentration. A significant effect of chromanol 293B was observed at the APD80 and APD90 (promoting triangulation of the AP shape), and these effects were greater when chromanol 293B was applied after E4031. In the absence of E4031, chromanol 293B had modest effects, and these effects were not significant at all the time points, maybe due to the low number of animals used. Moreover, ML-277 shortened the APD50, APD80, and APD90, and it partly restored the APD after applying E4031. These results show that the *I*_Ks_ is present in adult zebrafish cardiomyocytes and that the contribution of the *I*_Ks_ to the repolarization is greater when the *I*_Kr_ is compromised, suggesting that the *I*_Ks_ serves as a repolarization reserve in zebrafish as in humans. 

### 3.1. I_Kr_ and I_Ks_ Contribution to Zebrafish AP Repolarization

The small effect of chromanol 293B in prolonging the zebrafish heart APD shown here was previously described by Abramochlin et al. in 2018, who found that 100 µM chromanol 293B inhibited the *I*_Ks_ by only 36% in isolated zebrafish cardiomyocytes. The critical residues (T312, I337, F340) for the chromanol 293B binding site in human Kv7.1 are conserved in zebrafish Kv7.1 [19,33]. They found strong transcript expression of the KCNQ1 gene encoding the α-subunit of the Kv7.1 channel in the adult zebrafish atrium and ventricle. However, *KCNE1*, encoding the KCNE1 β-subunit, was expressed at 21- and 17-times lower levels in the ventricle and atrium, respectively, when compared to *KCNQ1* [19]. Moreover, for those currents not inhibited by E4031 in zebrafish cardiomyocytes (assumed to be mainly the *I*_Ks_), the activation kinetics resembled more the kinetics of Kv7.1 alone [19]. Abramochlin et al. therefore proposed that the low affinity of chromanol 293B they found in zebrafish cardiomyocytes might be due to the low expression of KCNE1 in zebrafish cardiomyocytes relative to human cardiomyocytes, because chromanol 293B is more potent in inhibiting human Kv7.1/KCNE1 than human Kv7.1. However, we found that the effect of chromanol 293B was greater on the zebrafish Kv7.1 than on the zebrafish Kv7.1/KCNE1 when recorded in *Xenopus laevis* oocytes: 10 µM chromanol 293B inhibited the zebrafish Kv7.1/KCNE1 expressed in *Xenopus laevis* oocytes by around 50%, and the same concentration almost inhibited 80% of the zebrafish Kv7.1 current. The inhibitory effects observed in zebrafish cardiomyocytes are similar when compared to the effects of chromanol 293B on human (IC50 ≈ 7 µM) and zebrafish Kv7.1/KCNE1 expressed in *Xenopus laevis* oocytes. In our study, the small effect observed in the zebrafish APD80 when 100 µM chromanol 293B and 10 µM ML-277 were applied compared with the results observed for zebrafish Kv7.1/KCNE1 and Kv7.1 (10 µM chromanol 293B and 1 µM ML-277) in *Xenopus laevis* oocytes suggests that the contribution of Kv7.1 or the Kv7.1/KCNE1 to the zebrafish action potential under resting conditions is small. Moreover, all the zebrafish Kv7.1/KCNE1 and Kv7.1 inhibitory and activating effects analyzed (Appendix A) in *Xenopus laevis* oocytes suggest that the contribution of Kv7.1 is higher than that of Kv7.1/KCNE1. Our results are consistent with the earlier study showing limited expression of KCNE1 in zebrafish cardiomyocytes [19] For example, we see a bigger effect in the zebrafish heart of ML-277 than the two PUFA compounds, just as ML-277 has bigger effects than the two PUFAs on the Kv7.1 channels expressed in oocytes. 

Understanding the role of the *I*_Ks_ and *I*_Kr_ is critical given the importance of these currents for the repolarization of the human heart. It should be noted that there are significant differences even among mammalian hearts when it comes to the activity of each ion channel. For instance, *I*_Kr_ inhibition in humans results in a significantly larger APD90 prolongation (80%) when compared to dogs (30%) and guinea pigs (20%), whereas *I*_Ks_ inhibition has almost no effect on human and dog APDs but severely prolongs the guinea pig APD [36,37]. Sympathetic activity of β-adrenergic receptors in guinea pig ventricular cells and human atrial and ventricular cells potentiates the *I*_Ks_ [36,38,39]. Furthermore, it has been demonstrated that although the *I*_Ks_ does not play a large role in the basal state of the human heart, it plays an important role in preventing APD prolongation when the human ventricular repolarization reserve is attenuated and the sympathetic tone is increased [37,40]. Here, we show that the addition of chromanol 293B after *I*_Kr_ inhibition by E4031 produced a further increase in the APD80 and APD90 (also observed in the triangulation analysis) in zebrafish hearts. Moreover, the effect of ML-277 (presumably through the *I*_Ks_ channels) on the APD90 was increased after applying E4031. These results suggest that the *I*_Ks_ may serve primarily as a reserve current for periods of stressed activity, and while under resting conditions, repolarization of the zebrafish heart is mostly determined by the *I*_Kr_. Zebrafish hearts therefore share a similar response to Kv7.1/KCNE1 and Kv11.1 inhibition as the human heart [37].

Our results demonstrated that when the *I*_Kr_ is compromised, the relevance of the *I*_Ks_ is significantly higher, suggesting a repolarization reserve mechanism for the *I*_Ks_ in zebrafish hearts.

### 3.2. Small Effects of Other Human I_Ks_ Inhibitors

The human *I*_Ks_ inhibitor JNJ303 did not prolong phase 3 and prolonged the APD25 and APD50 more than the APD80 and APD90 in the zebrafish optical AP recordings, suggesting that JNJ303 is not a good zebrafish *I*_Ks_ inhibitor. JNJ303 inhibition of human Kv7.1/KCNE1 has only been observed in the presence of the KCNE1 subunit. The JNJ303 binding site is described as in a fenestration in the pore domain formed at the interface of these two subunits, suggesting that JNJ303 is a gating modifier instead of a pore blocker [34]. We showed that JNJ303 has a very low affinity for zebrafish Kv7.1/KCNE1 expressed in *Xenopus laevis* oocytes (Appendix A). This suggests that there might be structural differences between human and zebrafish Kv7.1/KCNE1 and that the presence of the zebrafish KCNE1 is not enough to recapitulate the JNJ303 effect observed in human Kv7.1/KCNE1. Note also that although Kv7.1 is very conserved between human and zebrafish, KCNE1 is less conserved between these two species. The less conserved KCNE1 might play a role in the lesser effect of JNJ303 on zebrafish Kv7.1/KCNE1 than on human Kv7.1/KCNE1. The mechanism underlying the significant APD25 prolongation observed with JNJ303 on the zebrafish APD remains unclear. 

One possible explanation for the JNJ303-induced APD25 prolongation observed in the zebrafish AP could be that the outward potassium current (Ito) is inhibited by JNJ303. However, it has been suggested that because phase 1 is absent in the repolarization of the zebrafish AP, the Ito current might be small, or not present, in zebrafish cardiomyocytes [6,41]. There are no published data demonstrating that the Ito is present in zebrafish hearts and further investigation is required. The prolongation observed in the APD25 might be due to an increased activation or slowed inactivation in the T-type calcium channels. The T-type Ca^2+^ channels play a central role in generating and maintaining the plateau phase together with the L-type Ca^2+^ channels in zebrafish cardiomyocytes. In addition, the T-type Ca^2+^ channels are activated at more negative potentials compared to the L-type Ca^2+^ channels [41]. Hence, the fast and early activation of these channels might impair the phase 1 repolarization of zebrafish cardiomyocytes. Thus, an increase in the APD25 in the presence of JNJ303 might be caused by an increase in activation or a slowed inactivation of the T-type Ca^2+^ channels by JNJ303.

### 3.3. PUFA Analogues Slightly Shorten Zebrafish AP

Activators of Kv7.1/KCNE1 have been proposed as a suitable treatment for LQTS [42]. PUFA analogues have been shown as activators of human Kv7.1/KCNE1 and Kv7.1 by exerting dual effects to increase the *I*_Ks_ by (1) shifting the voltage dependence of activation toward negative potentials and (2) increasing the channel maximum conductance [31] (Figure 1B). However, PUFA analogues can also broadly modulate other cardiac voltage-gated ion channels. DHA-Gly and LIN-Gly (Figure 1A) were selected for this study due to their relatively high selectivity for human Kv7.1/KCNE1 and smaller effects on the human NaV and CaV channels [2]. DHA-Gly and LIN-Gly have been shown to shorten the QT interval and the APD in ex vivo drug-induced LQT2 guinea pig hearts [32,42]. DHA-Gly and LIN-Gly showed a moderate shortening effect on the zebrafish APD, consistent with a limited *I*_Ks_ or Kv7.1 contribution to repolarization under resting conditions. However, although DHA-Gly and LIN-Gly slightly shortened the zebrafish APD under rest conditions, they did not produce any shortening effect when LQTS was drug-induced with E4031 in the zebrafish heart. This could be explained by the fact that DHA-Gly and LIN-Gly exerted only modest effects on the zebrafish Kv7.1/KCNE1 and Kv7.1 expressed in *Xenopus laevis* oocytes.

In contrast to our results concerning the PUFA analogues DHA-Gly and Lin-Gly, we showed that the stronger Kv7.1 activator (and, most likely, more specific compound) ML-277 partly reversed the prolongation produced by E4031 in zebrafish hearts, suggesting that the PUFA analogues are not modulating Kv7.1/KCNE1 and Kv7.1 strongly enough in the zebrafish hearts to cause large effects on the APD. Similarly, PUFAs have been shown to have no or only small effects on cardiac arrhythmia in human clinical trials. Here, we have shown that a strong *I*_Ks_ activator, such as ML-277, has a significant effect on the APD in zebrafish hearts, suggesting that strong *I*_Ks_ activators might be good candidates for developing novel antiarrhythmic compounds. 

To conclude, our results show that zebrafish hearts share similarities with the human heart regarding the activation and contribution of the delayed rectifier potassium currents *I*_Ks_ and *I*_Kr_ involved in the cardiac AP repolarization [37]. We also demonstrate that effective *I*_Ks_ activators restore the APD in drug-induced LQT2 zebrafish hearts, which suggests that the zebrafish heart might be a good animal model to test the antiarrhythmic activity of the *I*_Ks_ activators, although there are some differences in the pharmacology of the zebrafish *I*_Ks_ compared to the human *I*_Ks_ that have to be taken into account in future drug development. 

## 4. Materials and Methods

### 4.1. Zebrafish Heart Isolation Procedure

All the studies in animals were conducted in accordance with the National Institutes of Health (NIH) Guide for the Care and Use of Laboratory Animals.

One-year-old male adult TAB wild-type zebrafish (n = 60) were purchased from the COX Science Centre of the University of Miami and kept under controlled conditions of temperature (between 26 and 28 °C) and pH (between pH 6.8 and 7.5). Ex vivo whole hearts were used to quantify the ventricular action potential (AP) using optical recordings. The zebrafish were euthanized via placement in an ice bath followed by transection of the spinal cord. Subsequently, the zebrafish were placed into a 60 mm Petri dish pre-coated with a layer of SYLGARD184 (Dow Corning, Midland, MI, USA) previously filled with chilled external solution. The external solution contained (in mM): 140 NaCl, 4 KCl, 1.8 CaCl_2_, 1 MgCl_2_, 10 glucose, and 10 HEPES. The pH was adjusted to 7.4. Then, the zebrafish were fixated using pins over the head and the caudal fin. Lifting from the top of the operculum, the pectoral fin was removed with gentle tensile force, and the surrounding dermal area was peeled off, exposing the heart to the external solution. To excise the heart from the body, the surrounding tissue was removed and the zebrafish heart was separated from the bulbus arteriosus. Once the heart was excised, it was incubated in pre-warmed external solution with the myosin II inhibitor (−)-blebbistatin (20 µM; Cayman Chemical, RRID:SCR_008945, Ann Arbor MI, USA) for 10 min to prevent contractions of the heart.

### 4.2. Action Potential Optical Recordings and Analysis

The AP parameters were analyzed by measuring the voltage across the plasma membrane on the ventricular surface of the zebrafish heart using the voltage-sensing dye CytoVolt1. The zebrafish heart was incubated with CytoVolt1 (also known as Di-4-ANBDQBS, 20 µM; Potentiometric Probes, Connecticut, CT, USA) for 60 min. The voltage-sensing dye was diluted in pre-warmed external solution plus 20 µM of (-)-blebbistatin (to avoid movement artifacts). The optical APs were recorded with external solution plus 20 µM of (-)-blebbistatin at a controlled temperature (28 °C). The AP optical recordings were made using a MacroFluo Leica Microscope coupled with a Leica APOZ6 zoom and a 5×/0.5 LWD objective. The data acquisition was performed using a photodiode (UDT sensors Inc. Hawthorne, CA) connected to an Axopatch 200B amplifier and an Axon Digidata 1550B digitizer (Molecular Devices, San Jose, CA, USA). The zebrafish hearts were paced at 1 Hz (30 Volt) with a two-electrode system and a Grass S 9 stimulator. Stocks of DHA-Gly (100 mM) and LIN-Gly (100 mM) were prepared in ethanol. The respective channel blocker and activator, E4031 (Alomone Labs, RRID:SCR_013570, Jerusalem, Israel), (-)-[3R,4S]-chromanol 293B and JNJ303 (Tocris Bioscience, RRID:SCR_003689, Bristol, UK), ML-277 (Sigma-Aldrich, RRID:SCR_008988, St. Louis, MO, USA), or PUFA analogues (DHA-Gly and LIN-Gly, Cayman Chemical, RRID:SCR_008945, Ann Arbor, MI, USA) were added accordingly and measurements were taken every 10 min.

The recordings were performed under control conditions (saline solution) and at three different times (after 10 min, 20 min, and 30 min) after applying each drug. From each recording, 10 AP were averaged. The APD was measured at 25, 50, 80, and 90 percent of repolarization to obtain APD25, APD50, APD80, and APD90, respectively (Figure 1C). APD25 and APD50 demonstrate the effect of the drugs on the early phases of the AP repolarization. APD80 reflects the entire repolarization process (phases 1, 2, and 3). APD90 captures the end of the repolarization. The absolute change in the APD (ΔAPD) between the drug and the control was analyzed in ΔAPD as the percentage of change (Figure 1C). To analyze phase 3 of the repolarization, the subtraction APD80–APD25 was compared between the control and drug/PUFA analogue conditions.

Moreover, we included the APD25/APD80 ratio as a quantitative measurement of the AP morphology. A smaller ratio reflects AP triangulation and a larger APD25/APD80 ratio represents a more squared AP shape. 

### 4.3. Molecular Biology

In vitro zebrafish *KCNQ1* (ZDB-GENE-061214-5) and *KCNE1* (ZDB-GENE-141104-1) cRNA were transcribed using an mMessage mMachine T7 RNA Transcription Kit (RRID:SCR_004098, Ambion, Austin, TX, USA). A total of 50 ng of cRNA was injected into defolliculated *Xenopus laevis* oocytes (EcoCyte Bioscience, RRID:SCR_014773, Austin, TX, USA). The oocytes were injected with *KCNQ1* cRNA and co-injected with *KCNQ1* cRNA and *KCNE1* cRNA at a ratio of 3:1, weight:weight, resulting in an excess of KCNE1 subunits compared to KCNQ1 subunits. After the cRNA injection, the oocytes were incubated in ND96 solution (96 mM NaCl, 2 mM KCl, 1 mM MgCl_2_, 1.8 mM CaCl_2_, 5 mM HEPES; pH = 7.5) for 2 to 3 days before the electrophysiological experiments.

### 4.4. Two-Electrode Voltage Clamp

The oocytes were recorded in a two-electrode voltage clamp configuration at room temperature. The recording pipettes were filled with 3 M KCl. The chamber was filled with ND96 solution. JNJ303 and chromanol 293B with different concentrations (0.01, 0.1, 1, 10 and 100 µM) in ND96 were perfused onto the oocytes. PUFA analogues with different concentrations (0.2, 0.7, 2, 7 and 20 µM) in ND96 were perfused onto the oocytes. From a holding potential of −80 mV followed by a hyperpolarizing prepulse to −140 mV, steps from −100 mV to +60 mV (in +20 mV increments) were applied to activate the channel followed by a tail voltage of −40 mV to obtain the tail current. 

A concentration–response relationship was analyzed with the Hill equation: I⁄I_0_ $=1+A/(1+{\left(\frac{Km}{x}\right)}^{n})$, where I is the current amplitude at 0 mV for each concentration, A is the relative increase in the current (ΔI/I_0_) caused by the drug, Km (IC50) is the apparent affinity of the drug, and x is the concentration and n is the Hill coefficient.

A conductance versus voltage (GV) curve was obtained by plotting the normalized tail currents versus different test pulses to determine the steady-state voltage dependence of the current activation. The tail currents were measured at −40 mV following the test pulses. Then, the GV curve was fitted with a single Boltzmann equation: G(V) = A_min_ + (A_max_ − A_min_)/(1 + exp((V − V_1⁄2_)/K)), where A_max_ and A_min_ are the maximum and minimum conductance, respectively, V_1⁄2_ is the voltage where 50% of the maximal conductance level is reached and K is the slope factor. Data were normalized between the A_max_ and A_min_ values of the fit.

### 4.5. Statistics

Shapiro–Wilk tests were performed in all the collected data to test for a normal distribution. Then, the statistical data significance was determined using a paired *t*-test, where *p* < 0.05 was considered statistically significant. Each time for each drug was compared with its control (same experiment without drug). These time-matched control recordings were performed to rule out any effects that were unrelated to the drugs, such as the rundown of channel currents due to the excision of the zebrafish heart. In addition, the average of the drug effect at different times was compared with its control (same experiment without drug). Data were represented as the mean ± standard error of the mean (SEM) and the number of experiments (n) for each set of experiments.

## 5. Study Limitations, Future Directions, and Clinical Relevance

To not use an unethically high number of animals, we have limited ourselves to the number of animals in this study. We agree that some experiments could have used more animals to reach significance. However, in cases where clear significance was not met, the results showed small effects that even if shown to be significant with additional animals, are most likely not physiologically important due to the small effects. Our group has recently developed PUFA analogues that are more effective *I*_Ks_ activators [43], and maybe these will be more effective in shortening the APD in zebrafish hearts in future studies than the DHA-Gly and Lin-Gly used here. Due to some pharmacological differences between the human and zebrafish *I*_Ks_, one would have to make further studies on, for example, IPSC-derived human cardiomyocytes, to validate some of our conclusions. However, we still think that zebrafish is an early model for proof of principle that *I*_Ks_ activators, such as ML-277, might be potential treatments for long QT syndrome. 

## Figures and Tables

**Figure 1 ijms-24-12092-f001:**
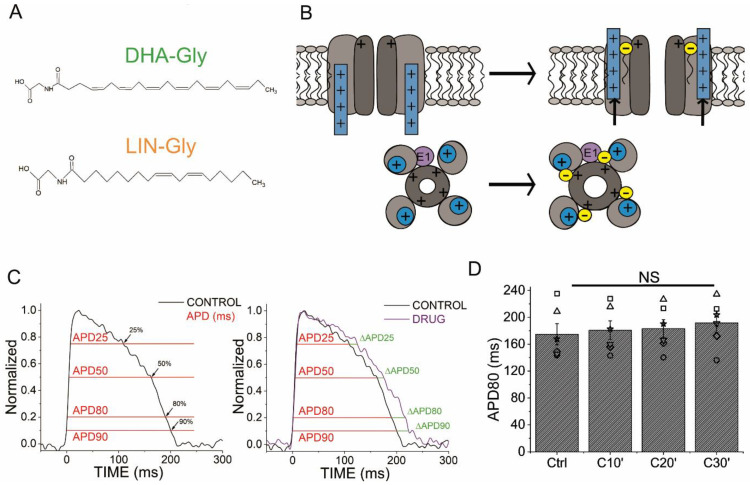
Schematic illustration of the topology of Kv7.1/KCNE1, the lipoelectric hypothesis and a zebrafish optical AP analysis diagram. (**A**) PUFA analogues’ molecular structure. (**B**) The lipoelectric hypothesis (previously shown in [2] but adapted to zebrafish Kv7.1/KCNE1 stoichiometry). Top, schematic illustration showing how the negatively charged head group of the PUFA analogue attracts the positively charged voltage sensor of the channel and activates the channel. Bottom, a second electrostatic effect of the negatively charged head group of the PUFA analogue with a positively charged residue located in the pore domain of the channels promotes the conductance increase in the channel. (**C**) Schematic of the optical AP analysis methodology. The left panel shows how the APD at different repolarization percentages was measured. The right panel shows how the change, ΔAPD (%), between the control conditions and the drug (in this example, chromanol 293B) was measured at different percentages of repolarization (APD25, APD50, APD80, and APD90). (**D**) APD80 (in ms) measured at different recording times (10, 20, and 30 min after the first control recording). Black bars show the control condition measured over time (C10′, C20′, and C30′) (mean ± SEM; NS: not significant). Each symbol represents data from a different experiment.

**Figure 2 ijms-24-12092-f002:**
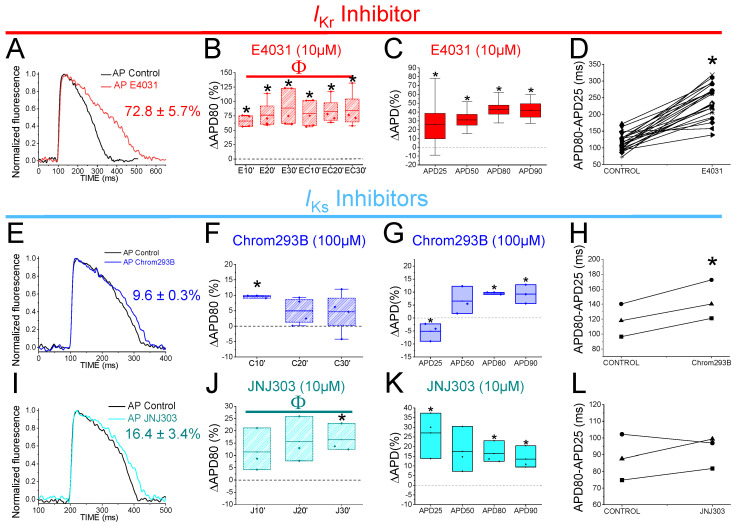
*I*_Kr_ and *I*_Ks_ inhibitors prolong zebrafish AP. (**A**) Representative zebrafish AP traces before (black) and after (red) applying E4031 (10 µM). (**B**) ΔAPD80 (%) after (red) applying E4031 (10 µM), as measured at different recording times (10, 20, and 30 min after application of the drug). (**C**) ΔAPD(%) produced by the drug measured for the APD25, APD50, APD80, and APD90. (**D**) Effect of the drug on phase 3 of the AP. APD80–APD25 under control conditions and after E4031 application. (**E**–**H**) The same as in (**A**–**D**) but for the *I*_Ks_ inhibitor chromanol 293B (100 µM). (**I**–**L**) The same as in (**A**–**D**) but for the *I*_Ks_ inhibitor JNJ303 (10 µM) (mean ± SEM; * *p* < 0.05 vs. control conditions, ^Φ^
*p* < 0.05 APD80 time data average vs. control condition). Each symbol in panels D, H, and L represent data from a different experiment.

**Figure 3 ijms-24-12092-f003:**
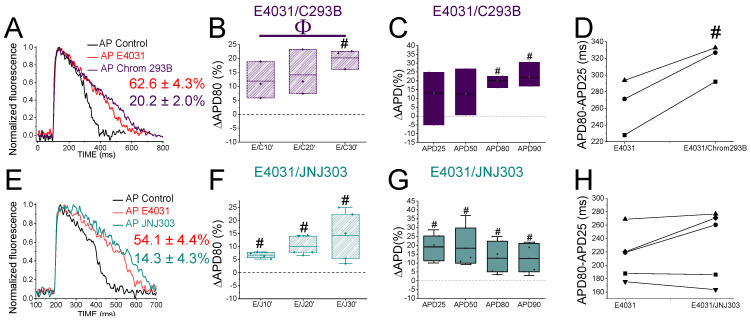
*I*_Ks_ inhibitors prolong more zebrafish AP after drug-induced LQT2. (**A**) Representative zebrafish AP traces before (black) and after applying E4031 (red, 10 µM), and E4031/chromanol 293B (10 µM/100 µM, dark purple). (**B**) ΔAPD80 (%) measured at different recording times vs. E4031 control condition (10, 20, and 30 min after application of E4031/chromanol 293B). (**C**) ΔAPD (%) produced by the combination of drugs measured for the APD25, APD50, APD80, and APD90. (**D**) Effect of the combination of drugs on phase 3 of the AP. APD80–APD25 in E4031 (control conditions) and after E4031/chromanol 293B application. (**E**–**H**) The same as in (**A**–**D**) but for the combination E4031/JNJ303 (10 µM/10 µM) (mean ± SEM; ^#^
*p* < 0.05 vs. E4031 condition, ^Φ^
*p* < 0.05 APD80 time data average vs. E4031 condition). Each symbol in panels D and H represent data from a different experiment.

**Figure 4 ijms-24-12092-f004:**
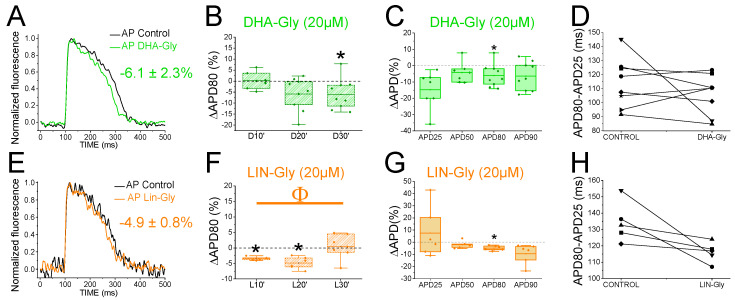
PUFA analogues have modest effects on zebrafish AP. (**A**) Representative zebrafish AP traces before (black) and after (light green) applying DHA-Gly (20 µM). (**B**) ΔAPD80 (%) after (light green) applying DHA-Gly (20 µM), as measured at different recording times (10, 20, and 30 min after application of the PUFA analogue). (**C**) ΔAPD (%) produced by the PUFA analogue measured for the APD25, APD50, APD80, and APD90. (**D**) Effect of the PUFA analogue on phase 3 of the AP. APD80–APD25 in control conditions and after DHA-Gly application. (**E**–**H**) The same as in (**A**–**D**) but for the PUFA analogue LIN-Gly (20 µM) (mean ± SEM; * *p* < 0.05 vs. control conditions, ^Φ^
*p* < 0.05 APD80 time data average vs. control condition). Each symbol in panels D and H represent data from a different experiment.

**Figure 5 ijms-24-12092-f005:**
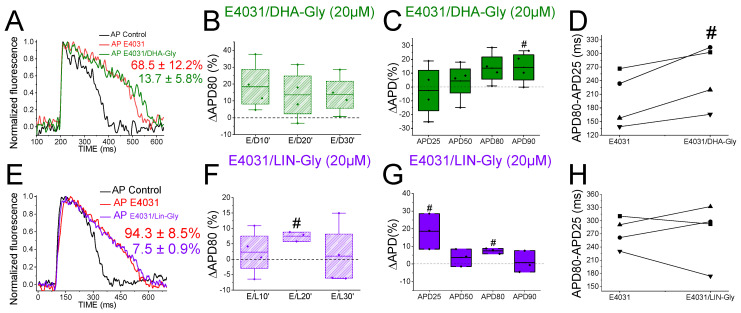
PUFA analogues do not shorten the zebrafish AP in LQT2 drug-induced conditions. (**A**) Representative zebrafish AP traces before (black) and after sequentially applying E4031 (10 µM, red) and E4031/DHA-Gly (10 µM/20 µM, dark green). (**B**) ΔAPD80 (%) measured at different recording times (10, 20, and 30 min after application of the combination drug/PUFA analogue). The E4031 recording, measured at 30 min, was used as the control condition. (**C**) ΔAPD (%) produced by the combination of drug/PUFA analogue measured for the APD25, APD50, APD80, and APD90. (**D**) Effect of the combination of drug/PUFA analogue on phase 3 of the AP. APD80–APD25 in E4031 (control conditions) and after E4031/DHA-Gly application. (**E**–**H**) The same as (**A**–**D**) but for the combination E4031/LIN-Gly (10 µM/20 µM) (mean ± SEM; ^#^
*p* <0.05 vs. E4031 condition). Each symbol in panels D and H represent data from a different experiment.

**Figure 6 ijms-24-12092-f006:**
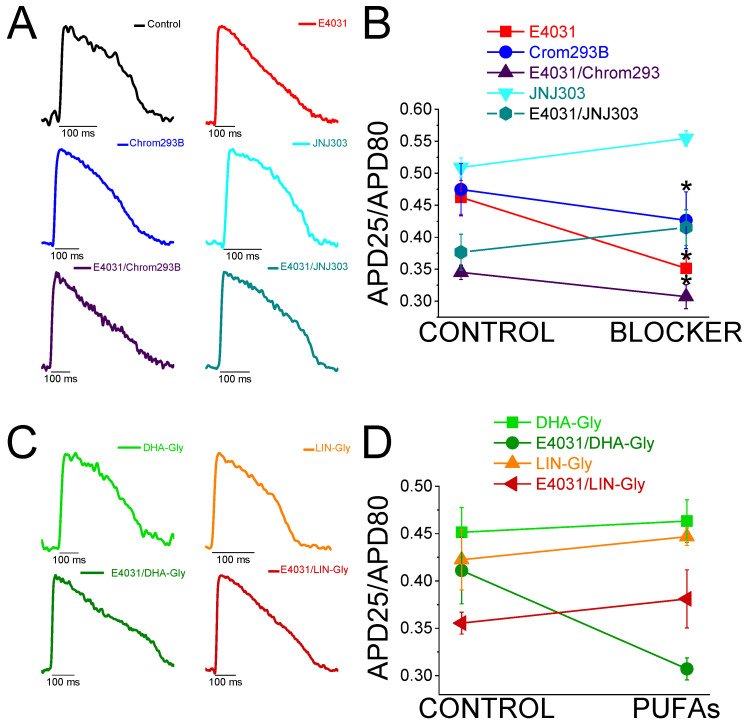
*I*_Kr_ and *I*_Ks_ inhibitors promote triangulation while *I*_Ks_ activators do not square the zebrafish AP shape. (**A**) Representative traces of the control conditions (black) and in the presence of the *I*_Kr_ and *I*_Ks_ inhibitors and the combination of inhibitors to show the qualitative effect of the triangulation of the different drug and drugs combination. (**B**) APD25/APD80 ratio in the control condition and after applying the drug for each drug or combination of drugs. (**C**) Representative traces in the presence of the PUFA analogues and the combination of E4031/PUFA analogues to show the qualitative effect of triangulation. (**D**) APD25/APD80 ratio in the control condition and after applying the PUFA analogues or E4031/PUFA analogues for each condition examined. For the drug combination conditions, CONTROL refers to the drug condition previously applied to the combination of drugs (mean ± SEM; * *p* < 0.05).

**Figure 7 ijms-24-12092-f007:**
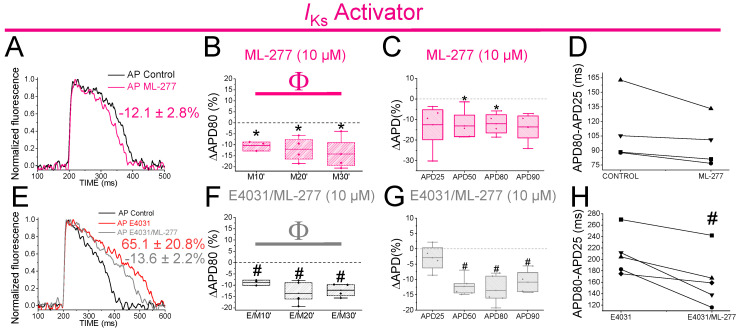
ML-277 shortens zebrafish AP before and after drug-induced LQT2. (**A**) Representative zebrafish AP traces before (black) and after (pink) applying ML-277 (10 µM). (**B**) ΔAPD80 (%) after applying ML-277 (10 µM), measured at different recording times (10, 20, and 30 min after application of ML-277). (**C**) ΔAPD(%) produced by ML-277 measured for the APD25, APD50, APD80, and APD90. (**D**) Effect of ML-277 on phase 3 of the AP. APD80–APD25 in the control conditions and after ML-277 application. (**E**) Representative zebrafish AP traces before (black) and after sequentially applying E4031 (10 µM, red) and E4031/ML-277 (10 µM/10 µM, grey). (**F**) ΔAPD80 (%) measured at different recording times (10, 20, and 30 min after application of the combination drug/ML-277). The E4031 recording, measured at 30 min, was used as the control condition. (**G**) ΔAPD (%) produced by the combination of drug/ML-277 measured for the APD25, APD50, APD80, and APD90. (**H**) Effect of the combination of drug/ML-277on phase 3 of the AP. APD80–APD25 in E4031 (control conditions) and after E4031/ML-277 application (mean ± SEM; * *p* < 0.05 vs. control conditions, ^#^
*p* < 0.05 vs. E4031 condition, ^Φ^
*p* < 0.05 APD80 time data average vs. its control condition; control condition or E4031 condition). Each symbol in panels D and H represent data from a different experiment.

**Table 1 ijms-24-12092-t001:** Average of the APD80 at different times. NS non significant.

	APD80 (ms)		APD80 (ms)	(n, *p*)
CTRL	174.9 ± 15.7	Long CTRL	185.2 ± 13.5	n = 6, NS, *p* > 0.05
E4031	332.2 ± 19.2	Long E4031	338.6 ± 17.8	n = 3, NS, *p* > 0.05
E4031 CTRL	217.6 ± 7.1	E4031	350.5 ± 11.4	n = 22, *p* < 0.001
C293B CTRL	225.9 ± 33.5	C293B	243.0 ± 39.8	n = 3, NS, *p* > 0.05
E4031	392.8 ± 14.0	E4031/C293B	463.5 ± 16.2	n = 3, *p* < 0.05
JNJ303 CTRL	180.2 ± 17.3	JNJ303	204.7 ± 13.5	n = 3, *p* < 0.05
E4031	325.6 ± 14.3	E4031/JNJ303	369.8 ± 28.7	n = 4, NS, *p* > 0.05
DHA-Gly CTRL	212.3 ± 9.1	DHA-Gly	206.1 ± 6.7	n = 7, NS, *p* > 0.05
LIN-Gly CTRL	210.8 ± 14.9	LIN-Gly	204.8 ± 15.1	n = 5, *p* < 0.05
E4031	325.3 ± 36.3	E4031/DHA-Gly	438.6 ± 54.7	n = 4, NS, *p* > 0.05
E4031	397.9 ± 14.6	E4031/LIN-Gly	433.0 ± 19.7	n = 3, NS, *p* > 0.05
ML-277 CTRL	236.4 ± 26.6	ML-277 CTRL	211.1 ± 22.3	n = 3, *p* < 0.05
E4031	301.5 ± 13.1	E4031/ML-277	272.2 ± 8.4	n = 3, *p* < 0.05

**Table 2 ijms-24-12092-t002:** Zebrafish APD measurements at different percentages of repolarization for the *I*_Kr_ and *I*_Ks_ inhibitors and *I*_Ks_ activator. NS non significant.

	ΔAPD25 (%)	(n, *p*)	ΔAPD50 (%)	(n, *p*)	ΔAPD80 (%)	(n, *p*)	ΔAPD90 (%)	(n, *p*)
E4031	48.3 ± 14.6	n = 26, *p* < 0.001	49.6 ± 4.8	n = 26, *p* < 0.001	72.8 ± 5.7	n = 26, *p* < 0.001	71.6 ± 5.8	n = 26, *p* < 0.001
C293B	−5.1 ± 2.0	n = 3, *p* < 0.05	6.5 ± 3.1	n = 3, NS, *p* > 0.05	9.6 ± 0.3	n = 3, *p* < 0.05	9.2 ± 2.1	n = 3, *p* < 0.01
JNJ303	27.1 ± 7.0	n = 3, *p* < 0.05	17.5 ± 6.8	n = 3, NS, *p* > 0.05	16.4 ± 3.4	n = 3, *p* < 0.01	13.6 ± 3.5	n = 3, *p* < 0.05
E4031/C293B	13.2 ± 9.2	n = 3, NS, *p* > 0.05	12.5 ± 7.6	n = 3, NS, *p* > 0.05	20.2 ± 2.0	n = 3, *p* < 0.01	22.0 ± 4.3	n = 3, *p* < 0.05
E4031/JNJ303	27.5 ± 10.5	n = 5, *p* < 0.05	20.2 ± 5.5	n = 5, *p* < 0.05	14.3 ± 4.3	n = 5, *p* < 0.05	14.0 ± 4.1	n = 5, *p* < 0.05
ML-277	−12.5 ± 6.0	n = 4, NS, *p* > 0.05	−13.1 ± 4.0	n = 4, *p* < 0.05	12.1 ± 2.3	n = 4, *p* < 0.05	−13.6 ± 3.8	n = 3, NS, *p* > 0.05
E4031/ML-277	3.9 ± 7.1	n = 5, NS, *p* > 0.05	−12.3 ± 1.5	n = 5, *p* < 0.01	−13.6 ± 2.2	n = 5, *p* < 0.01	−14.0 ± 3.5	n = 5, *p* < 0.05

**Table 3 ijms-24-12092-t003:** Zebrafish APD measurements at different percentages of repolarization for PUFA analogues. NS non significant.

	ΔAPD25 (%)	(n, *p*)	ΔAPD50 (%)	(n, *p*)	ΔAPD80 (%)	(n, *p*)	ΔAPD90 (%)	(n, *p*)
DHA-Gly	−0.7 ± 10.5	n = 9, NS, *p* > 0.05	−4.5 ± 3.4	n = 9, NS, *p* > 0.05	−6.1 ± 2.3	n = 9, *p* < 0.05	−6.4 ± 3.0	n = 9, NS, *p* > 0.05
LIN-Gly	7.6 ± 8.4	n = 6, NS, *p* > 0.05	−6.4 ± 4.5	n = 6, NS, *p* > 0.05	−4.9 ± 0.8	n = 6, *p* < 0.01	−9.4 ± 3.3	n = 6, NS, *p* > 0.05
E4031/DHA-Gly	−2.6 ± 9.5	n = 4, NS, *p* > 0.05	4.4 ± 6.9	n = 4, NS, *p* > 0.05	13.7 ± 5.8	n = 4, NS, *p* > 0.05	14.2 ± 5.8	n = 4, *p* < 0.05
E4031/LIN-Gly	18.6 ± 5.8	n = 3, *p* < 0.05	3.7 ± 2.9	n = 3, NS, *p* > 0.05	7.5 ± 0.9	n = 3, *p* < 0.05	0.8 ± 3.6	n = 3, NS, *p* > 0.05

**Table 4 ijms-24-12092-t004:** Triangulation (APD25/APD80 values). NS non significant.

	APD25/APD80		APD25/APD80	(n, *p*)
E4031 CTRL	0.48 ± 0.03	E4031	0.36 ± 0.02	n = 24, *p* < 0.001
C293B CTRL	0.47 ± 0.14	C293B	0.41 ± 0.06	n = 3, *p* < 0.01
E4031	0.35 ± 0.01	E4031/C293B	0.31 ± 0.02	n = 3, *p* < 0.05
JNJ303 CTRL	0.51 ± 0.02	JNJ303	0.55 ± 0.01	n = 3, NS, *p* > 0.05
E4031	0.38 ± 0.03	E4031/JNJ303	0.42 ± 0.03	n = 5, NS, *p* > 0.05
DHA-Gly CTRL	0.45 ± 0.02	DHA-Gly	0.46 ± 0.02	n = 9, NS, *p* > 0.05
LIN-Gly CTRL	0.41 ± 0.04	LIN-Gly	0.46 ± 0.01	n = 5, NS, *p* > 0.05
E4031	0.38 ± 0.11	E4031/DHA-Gly	0.33 ± 0.1	n = 3, NS, *p* > 0.05
E4031	0.34 ± 0.02	E4031/LIN-Gly	0.36 ± 0.04	n = 5, NS, *p* > 0.05
ML-277 CTRL	0.51 ± 0.03	ML-277 CTRL	0.51 ± 0.03	n = 4, NS, *p* > 0.05
E4031	0.37 ± 0.03	E4031/ML-277	0.44 ± 0.03	n = 5, NS, *p* > 0.05

## Data Availability

Data is available by reasonable request to the corresponding author.

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
