# Peer review of "Pharmacological Screening of Kv7.1 and Kv7.1/KCNE1 Activators as Potential Antiarrhythmic Drugs in the Zebrafish Heart"

_ijms, 2023, doi:10.3390/ijms241512092_

Round 1
Reviewer 1 Report
This study investigates the effects of PUFA analogues on IKs analogues on the cardiac action potential of zebrafish hearts. The results show that IKs is present in adult zebrafish cardiomyocytes and that the contribution of IKs to the repolarization is greater when IKr is compromised, suggesting that IKs serves as a repolarization reserve in zebrafish as in humans. The study results show that potent IKs activators can restore the action potential duration in drug-induced LQT2 in zebrafish heart. This study was well designed, and the study results were significant. Some comments are listed as below.
1. The number of experimental animals varies greatly among the groups. Some groups have too few animals.
2. This study suggested that there are some differences in the pharmacology of zebrafish IKs compared to human IKs. Whether this difference may have an impact on clinical application?
3. The typesetting of Table 4 in the article needs to be adjusted.
Author Response
This study investigates the effects of PUFA analogues on IKs analogues on the cardiac action potential of zebrafish hearts. The results show that IKs is present in adult zebrafish cardiomyocytes and that the contribution of IKs to the repolarization is greater when IKr is compromised, suggesting that IKs serves as a repolarization reserve in zebrafish as in humans. The study results show that potent IKs activators can restore the action potential duration in drug-induced LQT2 in zebrafish heart. This study was well designed, and the study results were significant. Some comments are listed as below.
- We thank the Reviewer for his/her praise of the study and for constructive critiques.
- The number of experimental animals varies greatly among the groups. Some groups have too few animals.
- To not use an unethical high number of animals, we have limited ourselves to the number of animals in this study. We agree that some experiments could have used more animals to reach a significance. However, in cases where clear significance were not met, the results showed small effects that, even if shown significant with additional animals, are most likely not important due to the small effects. We now also list this as Study Limitations in the Discussion.
- This study suggested that there are some differences in the pharmacology of zebrafish IKs compared to human IKs. Whether this difference may have an impact on clinical application?
- Yes, due to some pharmacological differences between human and zebrafish IKs, one would have to make further studies on, for example IPSC-derived human cardiomyocytes, to validate some of our conclusions. However, we still think that zebrafish is an early model for proof-of-principle that IKs activators might be potential treatments for Long QT Syndrome. We further discuss this in the Discussion of putative clinical relevance.
- The typesetting of Table 4 in the article needs to be adjusted.
- Thank you. The typesetting has been adjusted by the editor.
Reviewer 2 Report
This study aimed to investigate the role of the cardiac potassium current IKs in zebrafish hearts and its potential implications for Long QT syndrome (LQTS), a condition associated with ventricular arrhythmia and sudden cardiac death. The researchers used high-resolution all-optical electrophysiology on ex-vivo zebrafish hearts to assess the effects of IKs analogues on the cardiac action potential. The authors found that chromanol 293B, an IKs inhibitor, prolonged the action potential duration (APD) in the presence of E4031, an IKr inhibitor, mimicking drug-induced LQT2. The APD was also slightly prolonged in the absence of E4031, suggesting a limited role of IKs in ventricular repolarization under resting conditions. Furthermore, PUFA analogues, known as IKs activators, were found to slightly shorten the APD of the zebrafish heart. However, they failed to reverse the APD prolongation in drug-induced LQT2. In contrast, a more potent IKs activator, ML-277, partially reversed the APD prolongation in drug-induced LQT2 zebrafish hearts. This suggests that IKs may serve as a repolarization reserve in zebrafish hearts, similar to human hearts, and potent IKs activators could potentially restore the action potential duration in drug-induced LQT2. Overall, the study demonstrates the similarities between zebrafish and human cardiac electrophysiology and suggests that zebrafish hearts could serve as a valuable model for studying human cardiac diseases, including LQTS. The findings also highlight the potential of potent IKs activators as candidates for developing novel antiarrhythmic compounds.
There are several minor issues in the manuscript that needs to be addressed before considering it for publication.
Results:
3.1. Control Conditions: Authors used time-matched control conditions, but it does not explain the purpose of these controls. For someone who do not have expertise in such experiments, it would be difficult to interpret the results. It would be essential to provide more context on why time-matched controls were included in the study and how they contribute to the interpretation of the results.
Authors compared APD values between control conditions and drug or PUFA analogue conditions, it does not detail the specific drugs or PUFA analogues used in the study. Providing this information would be helpful for understanding the experimental interventions.
3.2 The authors mentioned that the results are consistent with previous studies (Reference 6), but it does not elaborate on the specific studies or how the current findings contribute to the existing knowledge. They did not discuss this in the discussion section also.
3.3. The passage points out that Chromanol 293B had a small but significant effect on APD80 at 10 minutes, but this effect was not maintained at 20 or 30 minutes after drug application. It's interesting to note that this small effect is consistent with a previous study in isolated zebrafish cardiomyocytes. However, authors did not discuss the potential reasons for the transient effect of Chromanol 293B at 10 minutes or provide further insight into the implications of this finding.
The passage reveals that JNJ303 caused a significant prolongation of APD80 after 30 minutes of application, but it unexpectedly affected APD25 more than APD80 or APD90. This observation raises questions about JNJ303's specificity for Kv7.1/KCNE1 channels and whether it may be influencing other ion channels involved in the zebrafish action potential. Authors should elaborate on these potential off-target effects or their significance in the context of the study.
The passage indicates that Chromanol 293B modified phase 3 of the action potential, as expected for an IKs inhibitor, but JNJ303 did not show a significant effect on phase 3. This difference suggests that JNJ303's impact on the zebrafish action potential might be different from that of Chromanol 293B, this discrepancy needs to be discussed
3.4 A small sample size (n = 3) for certain experimental conditions. While this is a limitation, it's understandable considering the complexities of zebrafish experiments. However, the authors should acknowledge this limitation and discuss how it might impact the reliability and generalizability of the findings.
The authors observed that Chromanol 293B had a larger effect on APD when IKr was blocked, suggesting that IKs plays a more crucial role in repolarization under these conditions. This finding raises questions about the potential compensatory mechanisms and interactions between IKr and IKs in zebrafish ventricles. The authors should discuss these mechanistic implications and provide insights into how their findings align with existing knowledge in cardiac electrophysiology.
3.6 Discussion of Selectivity: The authors showed the selectivity of JNJ303 and chromanol 293B for zebrafish Kv7.1/KCNE1 channels. However, it does not fully address the differences between the effects of these drugs on zebrafish Kv7.1/KCNE1 compared to human Kv7.1/KCNE1. The authors should discuss the potential reasons for the observed differences in drug sensitivity between the species and how these findings might impact the interpretation of the in vivo experiments.
Discussion:
The study confirms previous findings regarding the small effect of chromanol 293B on zebrafish heart APD. The authors should discuss potential reasons for the limited efficacy of chromanol 293B and how this might impact its potential use as an IKs inhibitor in zebrafish hearts.
The study provides interesting data regarding the differential effects of PUFA analogues and JNJ303 on zebrafish APD. The authors should explore the potential structural differences between human and zebrafish Kv7.1/KCNE1 that may contribute to the observed discrepancies in the response to these compounds.
The results suggest that PUFA analogues have only a modest effect on zebrafish APD under resting conditions and fail to reverse the prolonged AP when LQT2 is drug-induced. The authors should discuss the implications of these findings in terms of the potential use of PUFA analogues as antiarrhythmic agents.
The study highlights the importance of strong IKs activators, such as ML-277, in restoring APD in drug-induced LQT2 zebrafish hearts. The authors should elaborate on the significance of these findings and their potential relevance for drug development targeting IKs.
Study Limitations: The authors should acknowledge any limitations in the study design or methodology that may affect the interpretation of the results. Additionally, it would be helpful to discuss potential directions for future research based on the findings of this study.
Clinical Relevance: The authors should discuss the potential clinical implications of their findings, especially regarding the use of zebrafish hearts as a model for antiarrhythmic drug testing and the possible translation of results to human therapy.
Author Response
This study aimed to investigate the role of the cardiac potassium current IKs in zebrafish hearts and its potential implications for Long QT syndrome (LQTS), a condition associated with ventricular arrhythmia and sudden cardiac death. The researchers used high-resolution all-optical electrophysiology on ex-vivo zebrafish hearts to assess the effects of IKs analogues on the cardiac action potential. The authors found that chromanol 293B, an IKs inhibitor, prolonged the action potential duration (APD) in the presence of E4031, an IKr inhibitor, mimicking drug-induced LQT2. The APD was also slightly prolonged in the absence of E4031, suggesting a limited role of IKs in ventricular repolarization under resting conditions. Furthermore, PUFA analogues, known as IKs activators, were found to slightly shorten the APD of the zebrafish heart. However, they failed to reverse the APD prolongation in drug-induced LQT2. In contrast, a more potent IKs activator, ML-277, partially reversed the APD prolongation in drug-induced LQT2 zebrafish hearts. This suggests that IKs may serve as a repolarization reserve in zebrafish hearts, similar to human hearts, and potent IKs activators could potentially restore the action potential duration in drug-induced LQT2. Overall, the study demonstrates the similarities between zebrafish and human cardiac electrophysiology and suggests that zebrafish hearts could serve as a valuable model for studying human cardiac diseases, including LQTS. The findings also highlight the potential of potent IKs activators as candidates for developing novel antiarrhythmic compounds.
There are several minor issues in the manuscript that needs to be addressed before considering it for publication.
- We thank the reviewer for his/her comments and the constructive critiques.
Results:
3.1. Control Conditions: Authors used time-matched control conditions, but it does not explain the purpose of these controls. For someone who do not have expertise in such experiments, it would be difficult to interpret the results. It would be essential to provide more context on why time-matched controls were included in the study and how they contribute to the interpretation of the results.
- We have now included an explanation for the time-matched controls in the Methods and how they help in interpretation of the results.
Authors compared APD values between control conditions and drug or PUFA analogue conditions, it does not detail the specific drugs or PUFA analogues used in the study. Providing this information would be helpful for understanding the experimental interventions.
- Figure 1 was only meant to show the general idea of how we measure the effects of any drug (in this example it was chromanol). We now include the drug or PUFA analogues in each experiment.
3.2 The authors mentioned that the results are consistent with previous studies (Reference 6), but it does not elaborate on the specific studies or how the current findings contribute to the existing knowledge. They did not discuss this in the discussion section also.
- We now compare with previous studies and discuss our findings in relationship to these studies.
3.3. The passage points out that Chromanol 293B had a small but significant effect on APD80 at 10 minutes, but this effect was not maintained at 20 or 30 minutes after drug application. It's interesting to note that this small effect is consistent with a previous study in isolated zebrafish cardiomyocytes. However, authors did not discuss the potential reasons for the transient effect of Chromanol 293B at 10 minutes or provide further insight into the implications of this finding.
- We don’t know the reason for this transient effect. We attributed it to a small effect that might still be present at later points but due to the small effect it is hard to show significance without using an unethical high number of animals.
The passage reveals that JNJ303 caused a significant prolongation of APD80 after 30 minutes of application, but it unexpectedly affected APD25 more than APD80 or APD90. This observation raises questions about JNJ303's specificity for Kv7.1/KCNE1 channels and whether it may be influencing other ion channels involved in the zebrafish action potential. Authors should elaborate on these potential off-target effects or their significance in the context of the study.
- We now include a discussion about what potential side effects JNJ303 might have on other channels and the role of this in the Discussion.
The passage indicates that Chromanol 293B modified phase 3 of the action potential, as expected for an IKs inhibitor, but JNJ303 did not show a significant effect on phase 3. This difference suggests that JNJ303's impact on the zebrafish action potential might be different from that of Chromanol 293B, this discrepancy needs to be discussed.
- We now include a discussion about the different effects of Chromanol and JNJ303 and the role of this.
3.4 A small sample size (n = 3) for certain experimental conditions. While this is a limitation, it's understandable considering the complexities of zebrafish experiments. However, the authors should acknowledge this limitation and discuss how it might impact the reliability and generalizability of the findings.
- To not use an unethical number of animals, we have limited ourselves to the number of animals in this study if the effects were small. We agree that some experiments could have used more animals to reach a significance. However, in cases where clear significance were not met, the results showed small effects that, even if shown significant with additional animals, are most likely not important due to the small effects. We now also list this as Study Limitations in the Discussion.
The authors observed that Chromanol 293B had a larger effect on APD when IKr was blocked, suggesting that IKs plays a more crucial role in repolarization under these conditions. This finding raises questions about the potential compensatory mechanisms and interactions between IKr and IKs in zebrafish ventricles. The authors should discuss these mechanistic implications and provide insights into how their findings align with existing knowledge in cardiac electrophysiology.
- We now include a discussion of IKr and IKs interplay and refer to previous studies.
3.6 Discussion of Selectivity: The authors showed the selectivity of JNJ303 and chromanol 293B for zebrafish Kv7.1/KCNE1 channels. However, it does not fully address the differences between the effects of these drugs on zebrafish Kv7.1/KCNE1 compared to human Kv7.1/KCNE1. The authors should discuss the potential reasons for the observed differences in drug sensitivity between the species and how these findings might impact the interpretation of the in vivo experiments.
- We now include a discussion about what potential side effects JNJ303 might have on other channels and the role of this in the APD recordings. We also include a discussion about differences in structure of IKs channels in human and zebrafish and the potential role of these differences in the different effects or PUFAs and JNJ303.
Discussion:
The study confirms previous findings regarding the small effect of chromanol 293B on zebrafish heart APD. The authors should discuss potential reasons for the limited efficacy of chromanol 293B and how this might impact its potential use as an IKs inhibitor in zebrafish hearts.
- We now include a discussion about the small effects of Chromanol and the role of Chromanol in zebrafish studies.
The study provides interesting data regarding the differential effects of PUFA analogues and JNJ303 on zebrafish APD. The authors should explore the potential structural differences between human and zebrafish Kv7.1/KCNE1 that may contribute to the observed discrepancies in the response to these compounds.
- We now include a discussion about what potential side effects or PUFAs and JNJ303 might have on other channels and the role of this. We also include a discussion about differences in structure of IKs channels in human and zebrafish and the potential role of these differences in the different effects or PUFAs and JNJ303.
The results suggest that PUFA analogues have only a modest effect on zebrafish APD under resting conditions and fail to reverse the prolonged AP when LQT2 is drug-induced. The authors should discuss the implications of these findings in terms of the potential use of PUFA analogues as antiarrhythmic agents.
- We now include a discussion about the small effects of PUFA analogues on zebrafish APD and the role of PUFA analogues as antiarrhythmic agent.
The study highlights the importance of strong IKs activators, such as ML-277, in restoring APD in drug-induced LQT2 zebrafish hearts. The authors should elaborate on the significance of these findings and their potential relevance for drug development targeting IKs.
- We now include a discussion about the larger effects of ML-277 on zebrafish APD and potential of IKs activators as antiarrhythmic agent.
Study Limitations: The authors should acknowledge any limitations in the study design or methodology that may affect the interpretation of the results. Additionally, it would be helpful to discuss potential directions for future research based on the findings of this study.
- We now include a discussion about the limitations of this study and future directions.
Clinical Relevance: The authors should discuss the potential clinical implications of their findings, especially regarding the use of zebrafish hearts as a model for antiarrhythmic drug testing and the possible translation of results to human therapy.
- Due to some pharmacological differences between human and zebrafish IKs, one would have to make further studies on, for example IPSC-derived human cardiomyocytes, to validate some of our conclusions. However, we still think that zebrafish is an early model for proof-of-principle that IKs activators might be potential treatments for Long QT Syndrome. We further discuss this in the Discussion.